# Sustainable and Active Program—Development and Application of SAVING Methodology

**DOI:** 10.3390/ijerph19116803

**Published:** 2022-06-02

**Authors:** Marina Almeida-Silva, Ana Monteiro, Ana Rita Carvalho, Ana Marta Teixeira, Jéssica Moreira, David Tavares, Maria Teresa Tomás, Andreia Coelho, Vítor Manteigas

**Affiliations:** 1H&TRC-Health & Technology Research Center, Escola Superior de Tecnologia da Saúde (ESTeSL), Instituto Politécnico de Lisboa, 1900-096 Lisbon, Portugal; ana.monteiro@estesl.ipl.pt (A.M.); ana.ccarvalho.97@gmail.com (A.R.C.); a.marta97@hotmail.com (A.M.T.); moreira.jessica12@gmail.com (J.M.); david.tavares@estesl.ipl.pt (D.T.); teresa.tomas@estesl.ipl.pt (M.T.T.); andreiacoelhofs@gmail.com (A.C.); vitor.manteigas@estesl.ipl.pt (V.M.); 2Centro de Ciências e Tecnologias Nucleares, Instituto Superior Técnico, Universidade de Lisboa, 2695-066 Bobadela, Portugal; 3Centre for Research and Studies in Sociology (CIES-IUL), Instituto Universitário de Lisboa (ISCTE-IUL), 1649-026 Lisbon, Portugal; 4Centro Interdisciplinar de Estudo da Performance Humana (CIPER), Faculdade de Motricidade Humana, Universidade de Lisboa, 2780-052 Oeiras, Portugal

**Keywords:** aging, active, health, sustainability

## Abstract

The SAVING project aimed to create a sustainable and active aging program to promote the transition to sustainable aging in residential structures for the elderly (RSEs), developing research activities to apply the best strategies and good practices regarding the promotion of an active, healthy, and sustainable aging regarding social, economic, environmental, and pedagogic aspects. All this innovative methodology was built on a living-lab approach applied in one RSE, that was used as a case study. The results showed that the creation of the SAVING Brigade allowed not only increased reflection and mutual learning, but also created better conditions to face uncertainties and obstacles. Moreover, the use of indicators supported the basic themes and enabled comparison with other studies, between institutions or programs. Finally, the Action Plan acted as a tool for the development of previously defined strategies. It is possible to conclude that the breadth of the concept of quality of life encompasses the physical health of the individual, their psychological state, their social relationships, their perceptions, and the relationship with the characteristics of the context in which they are inserted. Therefore, active, sustainable, and healthy aging should be the goal.

## 1. Introduction

The world’s population is aging. Virtually in every country, the number of older people is increasing, which brings implications for nearly all sectors of society. Preparing for the economic and social shifts associated with an aging population is thus essential to fulfil the pledge of the 2030 Agenda for Sustainable Development that “no one will be left behind”.

Sustainable development is based on three pillars: economic growth, social inclusion, and environmental protection. These elements are interconnected, and all are crucial for the well-being of individuals and societies. In September 2015, a UN summit was held in New York (USA), which brought together world leaders (governments and citizens) to create a new global model to end poverty, promote prosperity and well-being for all, protect the environment, and combat climate change. They created the 2030 Agenda. It is a universal Agenda, to be implemented by all countries, based on 17 Sustainable Development Goals (SDGs) and 169 targets that the world must meet by 2030 to ensure well-being and prosperity for all people, with dignity and security in society [1].

This Agenda has provided a blueprint for shared prosperity in a sustainable world-covering such diverse but interconnected areas such as equitable access to education and quality health services; the creation of decent jobs; energy and environmental sustainability; conservation and management of the oceans; promoting effective institutions and stable societies; and combating inequality at all levels [2].

Trends in population aging are particularly relevant for the SDGs related to poverty eradication, the promotion of health, gender equality, employment, and sustainable human settlements, as well as promoting inclusive societies for older adults and reducing inequality within. Furthermore, it is more important than ever that governments design innovative policies and public services specifically targeted to older citizens, including policies addressing housing, employment, healthcare, infrastructure, and social protection, among others.

Since 1950, the world has been witnessing a reversal of the age pyramid, with a steady increase in the number of older adults worldwide. For instance, according to the United Nations in 2017, there were 962 million people aged over 60 years (at this time, the share of the population aged 60 years and over will increase from 1 billion in 2020 to 1.4 billion) [3], more than twice as large as in 1980 when there were 382 million older people worldwide. The number of elderly individuals is expected to double again by 2050 when it is projected to reach nearly 2.1 billion. Moreover, projections indicate that there will be more older persons aged 60 or over than adolescents and youth at ages 10–24 (2.1 billion versus 2.0 billion) [2].

The European aging rate in 2050 will be 263 per 100 children under the age of 15, maintaining the highest rate of aging compared to other continents. In addition, Europe has the highest percentage of older adult population in the world (in 2020, more than one fifth (20.6%) of the EU population was aged 65 and over) [4], with Africa being the continent with the lowest percentages of the population over 60 years of age.

Portugal, like other European countries, has been experiencing important demographic changes due to increased longevity and the percentage of the population aged 65 or older, as well as falling birth rates and the percentage of the population under 15 years [5]. The Portuguese aging rate in 2014 was higher than the European average, and Portugal is the 3rd most populous country in Europe, with 22.77% of the population aged 65 or over [6,7]. Additionally, in 2020, life expectancy reached 79.8 years for men and 85.3 years for women [8].

The reality that the world is, and will be, at least until 2050, creating enormous differences concerning the concerns of the older adult population has implications for numerous societal sectors. These implications have been accentuated by the decline in the active population as a result of the increase in the number of older adults, associated with a decrease in mortality, fertility, and birth rates over the decades. It has also been observed that the longevity index has been increasing in the last decades, from 41 to 48 in 10 years (2001–2010), with an increase in the longevity of the population. These indicators are considered positive because they result from the increase in the average life expectancy at the birth of the population (in Portugal, it has increased by around 20 years since 1950) [8], which was due to improvements at various levels, namely progress of developed societies in areas such as improvement of healthcare, universal access to healthcare, safety conditions at work, housing conditions, education, nutrition, environment, etc. [9,10]. Despite this achievement, longevity also represents a reduction in the functionality of biological organisms and morbidity, which often leads to disability, dependency, and extra healthcare needs. Furthermore, the increasing of the population also has implications for economics, education, and health, as well as for the environment [11].

One of the economic consequences of population aging is the loss of wealth generation. In the coming decades, the European Union will have a situation where it goes from 4 people of working age for each person aged 65 and over, to a ratio of only 2 persons of working age, for each person aged 65 or over [12]. This raises great economic as well as social problems since society ceases to be sustainable. Demographic factors related to the progressive aging of the population have effects on the increase of healthcare costs and on the greater difficulty of financing social policies. The percentage decrease in the active population and the increase in population dependent on state support in the form of subsidies and contributions contribute to both lower incomes and higher expenditure on social policies. The higher percentage of the population in the higher age groups implies greater use of the resources made available by the public services. Although 33% of the elderly in the EU27 felt in their full faculties to continue to exercise an economically paid profession after they reached retirement age, there are factors (such as the economic crisis that is felt in countries such as Portugal, Spain, and Greece) [13] that lead to an early retirement following agreements or redundancies which are followed by long-term unemployment. According to the United Nations [12], after 2008, there was a sharp decrease in the participation of the elderly population in economic activity, even after they reached retirement age and retired. This accentuates impoverishment, increasing the risk of poverty and social exclusion. The Portuguese sustainability index is 3.4 assets for each person aged 65 and over [14,15], a fact of concern for the state that must implement measures that counteract this effort rate of the elderly population. That said, we are witnessing a sense of disbelief, inactivity, and demoralization on the part of this population sector.

This conundrum constitutes a collective social and political awareness that is necessary to intervene, for instance, through the creation of new structures capable of responding to the new challenges. One example is the creation of specialized places to care for the older population. The number of these places, called Elderly Care Centers (ECC), increased by 49% in Portugal between 1998 and 2010 [16]. This type of structure addresses the specific needs during the aging process.

According to the United Nations [12], after 2008, there was an accentuated decrease in the participation of the elderly population in economic activities, which favored their exclusion from society. The fact that elderly people do not contribute productively to society results in the marginalization of this group. Therefore, it is necessary to implement measures for different realities.

“Active and sustainable aging” is defined as “the process of optimizing opportunities for health, participation, and security, to improve the quality of life as people age”, as well as the process of development and maintenance of functional capacity, which contributes to the well-being of the older adults [17]. The functional capacity should be evaluated by the relation between a person’s intrinsic capacities (physical and mental capacities) and their social context. In addition, it is important to note that WHO associates the term “active” not only with the ability to be physically active but “to increase healthy life expectancy as well as the quality of life for all people” who are in this process [17]. Moreover, active aging projects, programs, and policies that promote mental health and social relationships are as important as activities that improve physical health conditions.

As Wongsala and colleagues [18] stated, active aging applies to the whole community, and it results in the increased expectation of a healthy and quality life. To achieve the premise, it is necessary that individuals understand the potential for their physical, social, and mental well-being, therefore allowing and promoting active participation of older adults in economic, cultural, spiritual, and civic matters.

To fulfil this state, and not disregarding several projects of Green Care—a term that considers a set of therapeutic strategies that may include, for example, therapy using horticulture practices, rural animal-assisted therapy, and general therapy, based on spaces typically characterized as farms [19,20,21]—this project aims to create, develop, and validate a sustainable and active aging program named the SAVING Program, which is a novelty because it takes into account the main dimensions of sustainability and sustainable development (economic, social, and environmental dimensions), directly involving the institutionalized elderly in decision-making, being an active part of the whole process. The main objective is to improve the quality of life of the institutionalized elderly population. Through this program, older adults experience a sense of achievement at being able to have a say in the social, economic, and environmental management strategies of their Elderly Care Centers, ultimately steering them towards certification and the prestige which comes with being awarded a Sustainable Care Stamp (Sustainable and Active Aging Award). This reward will be awarded after the completion of 3/4 of each action plan and should serve as an incentive for the entire older community to continue to exceed sustainability goals.

## 2. Methodology

How can older people remain fully included in society after they retire? How can residential structures for the elderly, day care centers, or senior universities be attractive to active and non-active elderly people? How is it possible to improve the quality of life of the elderly?

The SAVING Program is an ideal way for care homes to embark on a meaningful path towards an effective and positive impact on the lives of old adults, their families, the ECC’s staff, and local authorities. Moreover, it is a massive opportunity not only to improve the ECC’s environmental and physical conditions but also to potentiate synergies with health, social, and academic professionals/institutions that might open an easy and free channel between old people and societal opportunities. The SAVING Program methodology is based on three pillars: (1) the creation of the SAVING Brigade; (2) the development of a database with the definition of several indicators regarding three strands (environmental, organization, and physical activity); and (3) the action plan. The goal is to potentiate the transition to sustainable and active aging and to build a solid foundation for the implementation of the program results after its completion.

### 2.1. Characterization of the Pilot Residential Structure for the Elderly

The creation of this program was driven by the living-lab methodology, which increases awareness and capacity building and fosters a communication framework between end-users and stakeholders. According to the existing literature, “living labs aren’t just an interesting topic that provides a multitude of research opportunities but also as a novel tool, methodology and design for practitioners to overcome a variety of challenges and needs” [22] Living-lab methodology encompasses several disciplines, promoting the fertilization of ideas and the capitalization of results of previous projects already validated through the creation of a framework of collaboration between scientific, technological, and business initiatives. It also creates a service that will significantly improve end-user capacity. This collaborative atmosphere enables the end-user to incorporate their own experience into the project’s products, while at the same time, the project results become apparent in their behavior.

For the implementation of the SAVING Program, the residential structure for the elderly “*Cooperativa de Solidariedade Social Os Amigos de Sempre*” was chosen, in São João da Talha, Loures, Portugal (Figure 1). This building is surrounded by a residential neighborhood, and it was built in 2009 with a total area of 2470 m^2^. Some facilities include, for instance, shared living rooms, recreational space for activities, a canteen, and a gym.

This residential structure for the elderly has a total of 71 institutionalized users and more than 50 employees including nurses, physical therapists, nursing assistants, and activity assistants, among others.

### 2.2. Definition of Base-Themes and KPI Generator

To manage, assess, and monitor the transition to a sustainable and active aging residential structure for the elderly, we used key performance indicators (KPIs) (Table 1). This mechanism made it possible to evaluate of a battery of sustainability indicators, selected according to the Sustainable Development Goals (SDGs) defined by the United Educational, Scientific, and Cultural Organization [23] being grouped into nine categories:(1)Energy: The methodology provides the characterization of the final consumption, associated energy costs, and production of on-site renewable energy. The evaluation was carried out considering the last year of elderly care center energy consumption.(2)Environmental quality: Environmental air quality contributes to raising awareness of the importance of a healthy indoor environment to improve quality of life [24] and to avoid the prevalence of sick building syndrome (SBS) symptoms [25]. Indoor air quality is directly affected by indoor occupancy, activity, equipment, materials, and outdoor air quality [26,27]. Indoor air quality (IAQ) was measured through on-site monitoring campaigns, together with the characterization of occupancy patterns, activities, and outdoor air pollutant concentrations. It involved the evaluation of a set of high priority pollutants, such as temperature, relative humidity, CO, CO_2_, PM10, PM2.5, total volatile organic compounds (TVOCs), fungi, and bacteria. Finally, indoor environment quality was assessed based on two criteria: measured pollutant concentrations exceeding the indoor air guideline values (IAGVs) defined by different standards and health organizations, and thermal comfort.(3)Waste management: The methodology requires information about the type of waste produced and recycled.(4)Water: The evaluation is based on the quantification of water consumption bills during the last year. The evaluation focuses on providing a scale of water consumption.(5)Mobility: The assessment of mobility was based on the availability of parking spaces for low-carbon transport modes, such as electric vehicles or bicycles, and public transport networks in elderly care center surroundings.(6)Green spaces: Green areas are critical for the health and well-being of people. The assessment was based on the identification and understanding of how outdoor spaces were managed. It was quantified by the green area surface, the number of plants and trees, and the use of chemicals and water.(7)Green deal: The assessment evaluates not only the purchasing of products and services and their environmental impact but also the evaluation of electric and electronic equipment with eco-labeling or energy labels and food consumption with an organic certificate.(8)Activities and organization: The assessment of this category was based on the number of cultural and social activities and the relationship between elders, relatives, and collaborators.(9)Physical activity and exercises: The assessment was based on the number and periodicity of physical activities and the type of such activities.

Appendix A presents the KPIs in detail. This table is categorically organized. Firstly, from (A) to (C) there are questions regarding data from the institution itself. From (D) to (Q) are indicators concerning the environmental strand which are divided into more in-depth questions on the topics: (D) to (M) energy category; (N) and (O) environmental quality category; and then, (P) waste management; (Q) water. Then, there are several questions regarding mobility (R), green spaces (S), green deal (T), activities and organization (U), and physical activities and exercises (V). These key performance indicators are tools that allow measuring the project’s performance level so that the objectives are achieved. The use of these indicators for later comparison with other studies, institutions, or programs will be favorable for the validation of these basic themes. There is a need to address aging in a comprehensive, holistic, and multidisciplinary way to identify the best trajectories for healthy aging.

KPIs must be defined to properly address the specificities of the elderly sector, and at the same time, it is a reference baseline allowing the performance comparison between different projects/initiatives. According to the WHO [3], active aging is a factor that improves people’s quality of life as they age. The characteristics of the buildings and the exterior have an important impact on the mobility, independence, and quality of life of the elderly [28]. Healthy aging can be influenced by the functional ability determined by intrinsic capacity, the environment, and the interactions between the two [29]. Two of the factors that make the subject more resistant to stressor attacks and psychiatric disorders are socio-environmental factors and exercise [29]. Social participation and social support are related to good health and well-being. Participation in leisure, social, and cultural subjects allows the elderly to continue to exercise their autonomy, and maintain or form supportive and caring relationships [28].

For these reasons, the main dimensions are active aging, social relationship, physical space, and environmental categories (energy, water, waste, indoor air quality). The KPI analysis can be top-down from the entire building to the smallest active equipment in it, and users can set the specific variables to obtain a benchmark according to their needs.

This module, through a mechanism of evaluation and anonymous ranking of ECCs, encouraged the transition to sustainable and active aging. The SAVING Index was created as the quality framework for the attribution of the SAVING Award.

### 2.3. Definition of Working Groups

To carry out the SAVING Program, it was necessary to define working groups, the objective was to assign tasks and responsibilities to each member.

The program involved the entire older adults’ community in the definition, implementation, and evaluation of measures conducted towards sustainable aging. Committing to a policy leading to sustainable aging in ECCs is an important first step toward ensuring success.

The working group was called the “SAVING Brigade” and it was composed of SAVING coordinators which comprise the older adult population, staff, students, teachers, and municipality. It was established that this group should meet a few times during the year to coordinate the project actions in the ECC. In all the reunions, the SAVING Brigade discussed new ideas, proposals, and ways to monitor and evaluate the actions proposed before. Moreover, the intention is to prioritize actions and set responsibilities for all the participants.

### 2.4. SAVING Index (Multi-Criteria Assessment)

The multi-criteria assessment is divided into nine categories, assessed by different key performance indicators presented in Table 1, which were obtained through technical audits, monitoring campaigns, and questionnaires, as illustrated in Figure 2.

Audits were based on technical inspections and checklists developed by the authors, to determine building characteristics, equipment, activities, behaviors, occupation pattern, and resource consumption of Care Centers buildings. Monitoring consisted of on-site measurements for the characterization of IAQ. Questionnaires were used to evaluate two specific categories: activities and organization, and physical activity and exercises.

## 3. Results and Discussion

In this following section, the SAVING Program will be introduced and explained. For the development of the database, 20 themes based on the main dimensions presented in Section 2.1—Characterization of the Pilot Residential Structure for the Elderly were selected. The checklist was organized into categories regarding three strands: environmental strand, activities and organization strand, and physical activity strand.

Table 1 presents the baselines for the subsequent application of the checklist. This table is categorically organized and can be seen in detail in the Appendix A. Firstly, from (A) to (C) there are questions regarding data from the institution itself. From (D) to (S) and (U) are indicators concerning the environmental strand, which are divided into more in-depth questions on the topics. The activities and organization component includes the indicator (T) and the physical activity component the indicator (V). Finally, there is a comment section so that some considerations and assessments can be noted during the application of this checklist.

The indicators presented above are subdivided into a series of questions that support the assessment instrument: the KPIs. These key performance indicators are tools that allow measuring the project’s performance level so that the objectives are achieved.

The use of these indicators for later comparison with other studies, institutions, or programs will be favorable for the validation of these basic themes.

As [30] argues, there is a need to address aging in a comprehensive, holistic, and multidisciplinary way to identify the best trajectories for healthy aging.

To understand this process, Figure 3 explains the International Classification of Functioning, Disability, and Health (ICF) [31] model.

This structure is based on the premise that health and social interactions are influenced by complex interactions between environmental factors, intrinsic factors related to the individual, the functions and structures of the organism itself, the activities and participation, as well as integration in society.

This IFC classification, as well as the SAVING Program, establishes and links environmental factors, physical activity, and social interactions as determinants for the well-being and health of a population. Therefore, environmental and social factors can facilitate or prevent active and sustainable aging, quality of life, and the physiological and psychological well-being of individuals.

The following are the basic themes that influence environmental factors and that are included in the checklist.

### 3.1. Resources Management

The diversity of forms of energy (water, light, and gas) used in a consumer installation (building) and the complexity of the different transformations (energy efficiency of equipment) that can intervene in the use of energy justify the need for strict energy management in an institution. An energy audit is an energy management methodology that assesses the energy efficiency of a building and allows to know the consumptions in detail and propose efficiency measures [32].

From the energy audits comes a document: The Building Energy Rating Certificate, which evaluates the energy efficiency of a property on a scale from F (very inefficient) to A + (very efficient), issued by independent qualified experts. In addition to this classification, the certificate contains information about the building’s characteristics: window insulation, ventilation, air conditioning, and hot water production and its effect on energy consumption, and indicates the improvement measures that can be taken to reduce consumption and improve occupant comfort and health.

Regarding the reduction of energy consumption, it is equally important to highlight natural lighting to the detriment of artificial electrical lighting. In addition to being a free source of energy, natural lighting is a strategic part of efficient energy management and allows to animate and create diversity in spaces. Additionally, there are currently recognized benefits for physical health that can neutralize seasonal affective disorder (SAD). However, excessive lighting can also be detrimental to comfort and even interrupt sleep [33].

### 3.2. Building Characteristics

Currently, there is a global trend to build sustainable and energy-efficient buildings. Windows, doors, and light facades with glass should contribute to ensuring thermal comfort conditions inside the buildings and reduce the energy needs for air conditioning when such equipment exists. These building components and the respective protective elements must contribute so that the interior surface temperature is close to the interior comfort temperature and thus limit phenomena of local discomfort, as well as the occurrence of surface condensations. The roof of a building, in addition to being the most exposed area, has the function of protecting the building from external influences such as sun, rain, heat, wind, and infiltrations. The roof also plays an important role in the durability of the materials applied to the building. Even though infiltrations are the most frequent cause for intervening on the roof, their repercussions, such as mold, are very harmful to health and precede the visible infiltration. The durability of the roof depends a lot on the choice of an adequate roofing system and the quality of its application. The facades are particularly susceptible to execution errors, which makes them the first elements of a building to highlight the consequences of poor planning or poor execution, particularly concerning energy efficiency. Consequently, it is highly relevant to consider the role and importance of thermal insulation on facades when constructing a building. With adequate thermal insulation, it is possible, on the one hand, to provide a desirable energy performance to the building and, on the other hand, improve interior comfort. In certain cases, it is even possible to improve the soundproofing, protecting the interior space from noise, without significant additional costs. This is especially convenient when the building is located next to streets or roads.

Moreover, the thermal design strategy should allow the creation of comfortable and stimulating thermal conditions that can exploit the most energy-efficient climatic conditions. The human body feels the thermal environment not only through air temperature but also under radiant conditions (sunlight), air movement (natural ventilation), and the conduction of heat through surface materials.

### 3.3. Environmental Management

Indoor air quality influences and affects, like the other environmental factors explained above, the well-being of the older adult population. Dimitroulopoulou [34] states that ventilation is recognized as an important component for a healthy home, since it is possible to associate poor/inadequate ventilation with indoor air pollution and, consequently, health problems. The indoor air quality and its impacts, particularly in situations where the occupants are most vulnerable (elderly population), can be quantified and solutions can be prescribed, both for the occupants and for the building itself (e.g., improved ventilation of spaces) [35]. Although the assessment of air quality is often made subjectively, associating it with odor, this assessment can be useful and indicate some type of risk to the health of the occupants; generally, indicators that present a health risk can only be measured analytically using proper devices. Comfort is understood as “a mental condition that expresses satisfaction” towards the environment and that incorporates qualitative and psychological considerations (expectation and control over the environment) and quantitative physical parameters (temperature, air circulation). However, to obtain a thermally more comfortable interior environment in buildings, it is necessary to use energy equipment, and consequently there is an increase in energy consumption. To promote sustainable aging, all these factors are considered to achieve comfort with the lowest possible energy consumption, seeking to obtain guarantees of economic and environmental sustainability.

In turn, waste planning and management are the objectives of current policies in the field of the environment, assuming a prominent role of a transversal nature due to the impact on the preservation of natural resources. In this context, waste should be seen as one of the most serious public and environmental health problems of contemporary life, and the consequences of inadequate production, handling, and disposal are directly and/or indirectly reflected in the health of the population and the survival of ecosystems.

Waste management has fundamental importance since it is in this way that waste is properly treated and sent to the most appropriate destination.

At the environmental level, the benefits translate into saving of the natural resources when waste is sent to an appropriate destination. The reintroduction in the production cycle of materials that have already been used and are now recycled avoids the use of natural resources. For this reason, great importance is given to integrated waste management from separation at source; this reintegration can lead to a more sustainable development of the economy and population, to avoid serious consequences for future generations [36].

Considering the increased awareness of the society for the preservation, protection, and improvement of the environment, this means that the institutions are also committed to responding to these concerns. To that end, they develop and implement procedures that enable them to achieve and demonstrate solid performance by controlling the environmental impact of their activities, products, and services.

An environmental management system enables a structured approach to establish environmental objectives, to achieve them, and to demonstrate that they have been achieved. The application of measures that promote environmental protection becomes an extremely important tool to control and prevent risks and impacts that may result from poor performance.

In this context, the chapter “Green Commitment” aims to evaluate information related to environmental management instruments that ensure a better environmental performance for organizations/institutions.

The importance of water resources is also highlighted. The notion of the importance of water in different areas of human life and the ecosystem that supports it is something that is commonly accepted [37]. Even so, water consumption has significant importance in energy consumption in nursing homes. The main factors that influence consumption in institutions are the flow, the duration of use, and the number of uses per day.

To reduce its use, strategies must be implemented for effective water management that should, in the first place, save on consumption, as it is certainly the most economical solution, not requiring large investments. In addition to the environmental factors previously mentioned, environmental barriers associated with inadequate accessibility of means of transport also influence the physical and psychological health of the elderly population [38]. Transport is a means that not only guarantees accessibility to other locations, but also functions as a mechanism for social inclusion. Difficulties in access to transport have implications on social life and leisure activities, reducing the quality of life of the population [39].

Illumination is also important given the needs of the older population. Efficient lighting is associated with a reduction in accidents (especially falls) and increases, above all, the confidence and safety of the elderly, providing a better quality of life [40].

The aging process affects the physical condition of individuals and one of the organs most affected is the vision. For this reason, according to the recommendations of the North American Lighting Engineering Society (IESNA), an older person needs a greater illuminance, about twice the necessity of most adult individuals.

Regarding green spaces, several studies [41,42,43] indicate that vegetation can affect behavior and translate into health gains. The presence of vegetation in the urban space, makes it a more pleasant place to live, work, and rest [44].

Natural environments with vegetation have tangible effects on aging, which stimulate activities of daily living, increasing satisfaction with the environment [45]. According to Sia [46], exposure to natural elements positively affects the health and well-being of the older population, as it generates a sense of connectivity to the natural environment due to the experiences and emotions related to the location, while promoting cognitive and sensory development.

In addition, green spaces can influence health by promoting physical activity and social contact [47], reducing stress. The existence of vegetation plays a very important role in promoting social interactions and contributes equally to the sense of community that is essential for social cohesion [48].

### 3.4. Physical Activity and Exercise

Although older adults decrease their levels of physical activity, health promotion should highlight the importance of physical activity levels and exercise, to slow the degree of loss of functional reserve which is closely related to the quality of life.

Langhammer and colleagues [49] affirm that “the main objective of the participation of the elderly in the physical activity programs is to promote the quality of life”. There is a lot of evidence on the benefits of regular exercise in the increment of an active life expectancy by slowing the development or progress of chronic diseases such as diabetes, osteoarthrosis, hypertension, cancer, etc., or other disabling conditions such as sarcopenia, frailty, balance disorders, and risk of falls among others [50,51]; and high mobility levels are connected to better functional capacity [51] and a better aerobic capacity correlates with functional performance [52].

Creating a culture of health is closely related to better physical and mental health and well-being and increasing physical activity levels contribute to empowering older adults in managing chronic conditions and maintaining functionality sustainably [53]. The “Decade for Healthy Aging 2021–2030 Program” is a decade of concerted, catalytic, sustained collaboration to improve the lives of older people, their families, and their communities and is the second action plan of the WHO Global strategy on aging and health [3]. Knowing that the prevalence of insufficient physical activity among adults aged 18+ years is in Portugal around 43.4% for both genders [54] is fundamental to increase not only the levels of physical activity but also the number of active adults with special attention to the increment in the number of active older adults. Langhammer and colleagues [49] show that the regular practice of physical activity, even started after the age of 65, contributes to greater longevity, improves physiological capacity, reduces the number of prescription drugs, brings psychological benefits, and improves the self-esteem of the elderly.

All older adults should undertake regular physical activity and should limit the amount of time spent being sedentary. Replacing sedentary time with physical activity of any intensity (including light intensity) provides health benefits [55] and the recommendations are to do at least 150–300 min of moderate-intensity exercise or at least 75–150 min of vigorous-intensity aerobic physical activity, or an equivalent combination along with muscle-strengthening activities at a moderate or greater intensity that involve all major muscle groups on two or more days a week [56].

### 3.5. Activities and Organization

Sustainable and active aging in elderly care centers is closely related both to the activities promoted by these organizations and to their internal organization. This implies the existence of qualified human resources that guarantee a wide and diversified set of cultural and leisure activities appropriate to the social profile, characteristics, and needs of users, taking place both inside and outside the centers. Therefore, to constantly improve the well-being of the elderly living in the centers, a sustainable and active program assumes the regular monitoring and evaluation of the activities implemented, as well as the general functioning of the organization, through a set of indicators, such as, among others, the existence of a formal activities plan, amount and diversity of activities developed, frequency, duration, and diversity of tours abroad, a periodic variation of activities inside and outside the center, information system for user’s family members, scheduling of the visits, a system of complaints, the relationship between employees and users, adequacy of the qualifications of employees to the tasks and functions performed, coherence of the performance assessment system, staff stability, adequacy, frequency, periodicity, and coverage of training courses for employees.

### 3.6. SAVING Index

The SAVING Index was implemented and tested in the selected residential structure for the elderly. Based on the local characteristics for each studied theme, it was possible to find a final result, which is presented in Figure 4. It was quite clear that the better results were associated with the activities and organization. The maximum value given in the “activities and organization” theme is because the selected residential structure for the elderly has a formal program of cultural and leisure activities, an internal evaluation system, an external evaluation system, annual activity reports, and annual activity plans. Green spaces and mobility showed scores higher than 4, mainly due to the type of garden and its sustainable treatment and due to public transportation systems and local parking existing nearby, respectively.

With 3.4 and 3 scores, respectively, there were the waste management and the physical activity and exercises. Although the residential structure for the elderly does the waste separation, it does not do its measurement. This means that it is not possible to assess the evaluation regarding waste production. Regarding the physical activity and exercise theme, there are several activity programs. However, none of them is intergenerational and there is a lack of therapeutical routes.

At the bottom of the index, there is environmental quality, water, green deal, and energy. Environmental quality is directly related to the importance and quality of indoor environmental quality, taking into consideration the thermal comfort and indoor air pollutants. Regarding the water theme, there are no measures that may reduce its consumption, such as the type of water supply or specific devices that could reduce the water flux. The residential structure for the elderly in the study already has some worries about the products that it procures, considering mainly local suppliers. Nevertheless, they do not have any environmental management system implemented nor is the number of electronic devices, at least, Class A. The latter has a direct impact on the energy theme, which presented a score of 1.3, the lowest one. It is still necessary to bet on LED lighting and to improve the air conditioning devices to be able to reduce the air conditioning hours.

## 4. Conclusions

The SAVING project aimed to create a sustainable and active aging program to promote the transition to sustainable aging in residential structures for the elderly, developing research activities to apply the best strategies and good practices regarding the promotion of active, healthy, and sustainable aging relative social, economic, environmental, and pedagogic aspects. To manage, assess, and monitor the transition to a sustainable and active aging residential structure for the elderly, key performance indicators were used, grouped into nine categories: energy, environmental quality, waste management, water, mobility, green spaces, green deal, activities and organization, and physical activity and exercises. The SAVING Index was implemented and tested in one selected residential structure for the elderly and the consistency of the indicators was confirmed.

Based on the implementation of this approach, not only the index as it is but also involving the whole internal community into the process, it was clear the potential improvement that can be done. Moreover, for the capabilities of the elderly, it could be seen as a possibility to be more active and involved in the local strategies and decisions. The application of the index in the pilot residential structure for the elderly allowed to optimize the methodology, to identify and to correct its weaknesses.

This methodology makes it possible to evaluate/characterize the various conditions (building and surroundings) of the residential structure for the elderly, thus supporting places that promote the health and well-being of their occupants.

The application of this methodology in more residential structures for the elderly is recommended, promoting the deep involvement of the elderly and the transition to a sustainable and active aging in residential homes.

## Figures and Tables

**Figure 1 ijerph-19-06803-f001:**
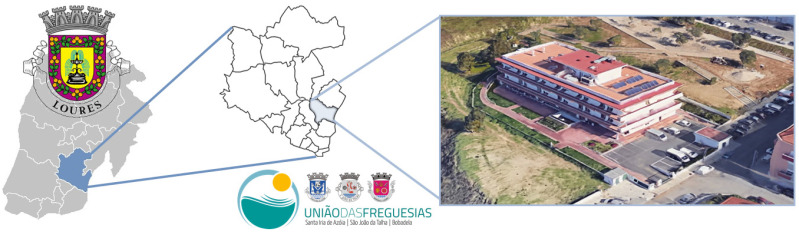
Geographical location of the residential structure for the elderly “*Cooperativa de Solidariedade Social Os Amigos de Sempre*”.

**Figure 2 ijerph-19-06803-f002:**
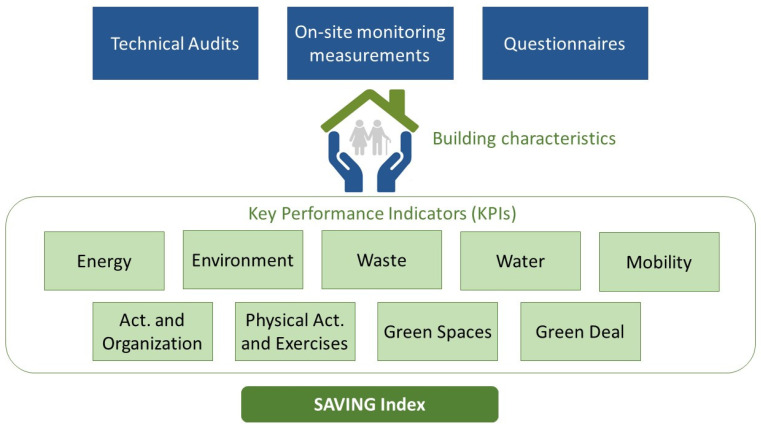
Overview of the multi-criteria assessment methodology.

**Figure 3 ijerph-19-06803-f003:**
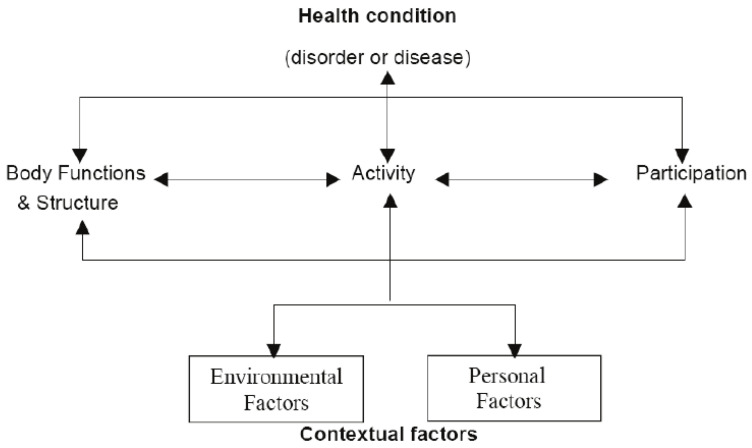
International Classification of Functioning, Disability, and Health (ICF) model [31].

**Figure 4 ijerph-19-06803-f004:**
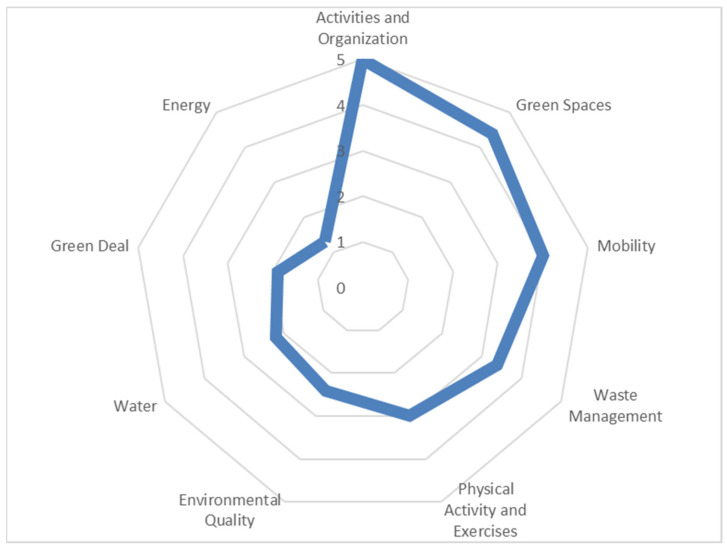
Final SAVING Index from pilot selected residential structure for the elderly.

**Table 1 ijerph-19-06803-t001:** Base-themes of SAVING’s checklist.

(A)	Administrative Data
(B)	Physical characteristics
(C)	Building usage
(D)	Energy consumption
(E)	Energy production
(F)	Lighting
(G)	Heating
(H)	Cooling
(I)	Ventilation
(J)	Energy measurement
(K)	Power management
(L)	Energy audits
(M)	Building features
(N)	Comfort
(O)	Indoor air quality
(P)	Waste management
(Q)	Water
(R)	Mobility
(S)	Green spaces
(T)	Activity and organization
(U)	Green deal
(V)	Physical activity and exercises
	Comments

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
