# Peer review of "Sustainable and Active Program—Development and Application of SAVING Methodology"

_ijerph, 2022, doi:10.3390/ijerph19116803_

Round 1

Reviewer 1 Report

Journal: IJERPH (ISSN 1660-4601)
Manuscript ID: ijerph-1727946
Type: Article
Number of Pages: 27

Title: Sustainable And Active Program – Development and Application of SAVING Methodology

Dear Authors,

It has been for me a great honour, as well as a pleasantly challenging activity, to review the article entitled ”Sustainable And Active Program – Development and Application of SAVING Methodology.

Overall, the article is interesting and easy to read. However, I suggest that the Authors introduce a few corrections (given below).

In my opinion, the Introduction chapter well introduces potential readers to the topics discussed by the Authors. It is based on well-chosen literature. However, the novelty of the article and what research gap it fills should be more clearly marked. In addition, it would be worth at least a few words to refer to the recently widely discussed issue of green care and its role in creating active aging, e.g.
DOI:10.3390/ani7040031 (Green Care: A Review of the Benefits and Potential of Animal-Assisted Care Farming Globally and in Rural America)
DOI 10.22616/ESRD.2018.148 (The Development of Green Care in Poland)
DOI:10.1016/j.jamda.2016.10.013 (Green Care Farms as Innovative Nursing Homes, Promoting Activities and Social Interaction for People With Dementia)

METHODOLOGY

This chapter is written in a clear and understandable way. It is logically divided into four following subsections. Figures 1 and 2 are a good illustration.

Technical error to be corrected on page 6:
“The multi-criteria assessment is divided into nine categories, assessed by different key performance indicators presented in Table 1, which were obtained through technical audits, monitoring campaigns, and questionnaires, as illustrated in Error! Reference source not found.

RESULTS AND DISCUSSION

This chapter presents the results of the research obtained in an understandable way. It is logically divided into six following subsections and it is well illustrated by 1 table and 2 figures. The research results were correctly discussed with other publications available on this subject.

Technical error to be corrected on page 12: Error! Reference source not found

Conclusions

The conclusions are clear. It is worth emphasizing the utilitarian nature of the study and the possibility of applying this solution in other similar cases.

I don't feel competent to comment on linguistic correctness as English is not my mother tongue. I can only say that the text reads well and the article has a good chance of attracting potential readers. I wish the Authors good luck.

Author Response

Dear Reviewer

Thank you so much fr all your kind words.

Based on your comments we had revised the article:

  1. In the introduction we had considered the very suitable articles that you had suggested and we believe that we'd improved the introduction accordingly.
  2. In both Methodology and Discussion there were two technical errors. They were both corrected accordingly.

I hope that you could accept our revision.

Many regards.

Reviewer 2 Report

Dear authors thank you for choosing this journal.  The article is very interesting and deals with a current issue in the contemporary scenario: defining sustainable strategies for quality population aging outside the pure medical field. The relationship you have defined in the study between human being and the physical and psychological state, in relation to the concept of "activity" even within the physical context is very important. If it were possible, consistent with privacy, I would suggest that you include some photos inside the case study so as to better link the physical space to the functional activities and users.

Author Response

Dear reviewer,

Thank you so much for your comments and positove feedback.

Unfortunantly, we are not allowed to add any further info about the Elderly Care Center in study. Perhaps in further researchers we could try to consider it.

Many regards